# Association of predicted lean body mass and fat mass with prognosis in patients with heart failure preserved ejection fraction

Wen Su[1], Xiaopu Wang[1,2], Qian wang[3], Junyu Pei[1]*, Zhenfei Fang[1]*

1 Department of Cardiovascular Medicine, The Second Xiangya Hospital, Central South University, Changsha, Hunan, China, 2 The Libin Cardiovascular Institute of Alberta, Cumming school of Medicine, The University of Calgary, Calgary, Alberta, Canada, 3 Department of biostatistics and data science, University of Texas Health Science Center, Houston, Texas, United States of America

* peijunyu@csu.edu.cn (JP); fangzhenfei@csu.edu.cn (ZF)

## Abstract

### Background

Previous studies have shown that body composition influences the prognosis of heart failure patients; however, the prognostic value of lean body mass and fat mass in patients with heart failure with preserved ejection fraction (HFpEF) remains unclear. This study aimed to investigate the association of lean body mass index and fat mass index with the prognostic outcomes in patients with HFpEF.

### Methods

We performed a post hoc analysis of data from the Treatment of Preserved Cardiac Function Heart Failure with an Aldosterone Antagonist (TOPCAT) trial to assess the relationship between Lean BMI and FMI with the adverse events. The primary endpoint was defined as a composite of cardiovascular death, aborted cardiac arrest, or heart failure hospitalization.

### Results

A total of 3,320 patients were included, with a median follow-up of 3.3 years. Among them, 624 occurred a primary endpoint event, and 503 died. Lean BMI was not associated with the primary endpoint (HR 0.98, 95% CI 0.82–1.16) but was found to reduce the risk of all-cause mortality (HR 0.74, 95% CI 0.61–0.91). In contrast, FMI was associated with an increased risk of both the primary endpoint (HR 1.29, 95% CI 1.09–1.55) and all-cause mortality (HR 1.46, 95% CI 1.19–1.79).

**Data availability statement:** All relevant data are within the paper and its Supporting Information files. The present study used data from the TOPCAT Trial obtained from the National Heart, Lung, and Blood Institute (https://biolincc.nhlbi.nih.gov/studies/topcat/). (NCT00094302) Detailed information on the study design, institutional approval, and primary outcomes had been previously reported.

**Funding:** This study was funded by two major projects from Hunan Province, China: 2021SK2004; 2021SK1040 awarded to Z.F.; the funder had no role in study design, data collection and analysis, decision to publish, or manuscript preparation.

**Competing interests:** The authors have declared that no competing interests exist.

**Abbreviations:** HFpEF, Heart failure with preserved ejection fraction; HFrEF, heart failure with reduced ejection fraction; Lean BMI, lean body mass index; FMI, fat mass index; TOPCAT, Treatment of Preserved Cardiac Function Heart Failure with an Aldosterone Antagonist; BMI, body mass index; DXA, dual-energy X-ray absorptiometry; CAD, Coronary heart disease; COPD, Chronic obstructive pulmonary disease; DM, diabetes; AF, Atrial fibrillation.

## Conclusion

In patients with HFpEF, a higher FMI was strongly associated with increased risks of both the primary endpoint and all-cause mortality, while an elevated LBMI was associated with a reduced risk of mortality.

## Introduction

Heart failure with preserved ejection fraction (HFpEF) is increasingly recognized as a major subtype of heart failure, with its prevalence rising in recent years [1,2]. Obesity is common in HFpEF, especially in the elderly, and is considered an important pathogenic factor [3,4]. Furthermore, reduced lean body mass or skeletal muscle mass has been linked to the onset and adverse prognosis of HFpEF [5]. Despite this, most studies have used body mass index (BMI) as an indicator of obesity to assess physical status, although BMI does not differentiate between lean body mass and fat mass [6]. Obesity is by definition characterized by a high fat mass, but the classic obesity phenotype is also characterized by either unchanged or increased lean body mass [7]. Individuals with the same BMI may have substantial variation in their body composition, and changes in both muscle mass and fat mass may affect outcomes in patients with heart failure [3,8–10].

However, HFpEF and heart failure with reduced ejection fraction (HFrEF) exhibit distinct differences in body composition, and alterations in lean body mass or fat mass may exert varying effects across different heart failure subtypes [7,11]. Thus, conclusions from previous studies focusing on general heart failure populations might not be directly applicable to HFpEF patients. Therefore, it is crucial to elucidate the role of lean body mass and fat mass in HFpEF to provide novel insights for clinical practice.

In the Treatment of Preserved Cardiac Function Heart Failure with an Aldosterone Antagonist (TOPCAT) trial [12], prior secondary analyses based solely on obesity classification failed to demonstrate a significant association between obesity and primary outcomes or all-cause mortality. The primary endpoint was defined consistent with the TOPCAT study, with the primary outcome comprising a composite of cardiovascular death, aborted cardiac arrest, or hospitalization due to heart failure. In this study, we employed previously validated anthropometric models to estimate body composition [12]. We evaluated the associations of estimated fat mass, lean body mass, and the risk of primary endpoint events and all-cause mortality in patients with HFpEF.

## Methods

### Study participants and data collection

The present study used data from the TOPCAT Trial obtained from the National Heart, Lung, and Blood Institute (https://biolincc.nhlbi.nih.gov/studies/topcat/) (NCT00094302). Detailed information on the study design, institutional approval, and primary outcomes had been previously reported [12]. The protocol was approved

by an ethics committee or institutional review board at each participating site. All participants provided written informed consent following established guidelines for the protection of human subjects. This study enrolled 3,445 patients aged ≥50 years with symptomatic heart failure and a left ventricular ejection fraction of at least 45% to evaluate whether spironolactone could improve outcomes in HFpEF. The median follow-up was 3.3 years, during which 671 participants experienced at least one primary outcome event, 526 patients died. However, no significant differences were observed between the treatment groups in time to all-cause mortality or first hospitalization [12]. Patients with missing values in the relevant data were excluded from the analysis cohort.

## Exposure variables and study outcome

We estimated lean body mass and fat mass using prediction equations derived from the NHANES survey, which included dual-energy X-ray absorptiometry (DXA) measurements of 7,531 men and 6,534 women [13]. These equations demonstrated high predictive accuracy for lean body mass and fat mass across different subgroups (e.g., age, BMI, disease status) with minimal bias. The equations, detailed in S1 Table in S1 File, incorporate age, sex, race, height (cm), weight (kg), and waist circumference (cm). Given that lean body mass and fat mass are related to height and can vary substantially among different body types, we calculated lean body mass index (Lean BMI) and fat mass index (FMI) by dividing lean body mass and fat mass (kg) by height squared ($m^2$) [10,13]. These indices were used in the statistical analyses.

Participants underwent standardized interviews, physical examinations, and laboratory tests at baseline, during follow-up, and at the end of the trial, following previously published protocols [14].

Coronary artery disease (CAD) was defined as a history of myocardial infarction, coronary artery bypass grafting, percutaneous coronary intervention, or angina. Smoking status was categorized as "never or former smoker" and "current smoker" (within the past 30 days). The primary endpoint of this study was consistent with the original TOPCAT trial, defined as a composite of cardiovascular death, aborted cardiac arrest, or hospitalization for heart failure, while all-cause mortality was considered a secondary endpoint [12,14]. The selection of risk factors was based on previous studies and clinical evidence.

## Statistical analysis

Baseline characteristics were presented as frequencies and percentages for categorical variables, and as means with standard deviations (SD) or interquartile ranges (IQR) for continuous variables, depending on the data distribution (assessed using normal Q-Q plots). Categorical variables were compared using the χ2 test, while continuous variables were compared using analysis of variance (ANOVA) or the Mann-Whitney U test, as appropriate.

Since our preliminary evaluation indicated that the relationship between Lean BMI and FMI and the study-defined endpoints was not entirely linear, we treated these variables as both continuous and categorical measures. Lean BMI and FMI were categorized into sex-specific quartiles. The median of each quartile was used as a continuous variable entered the Cox proportional hazards model for trend analysis.

We constructed Cox proportional hazards regression models adjusted for potential confounding variables to evaluate the association between Lean BMI and FMI with primary endpoint and all-cause mortality. We calculated hazard ratio (HR) for sex-specific quartiles, using the first quartile as the reference. Four models were employed: Model 1 (unadjusted); Model 2 (adjusted for age, sex, and race); Model 3 (Model 2 plus adjustments for spironolactone treatment, history of CAD, hyperlipidemia, hypertension, smoking, diabetes, and chronic obstructive pulmonary disease [COPD]). To assess the independent association between Lean BMI and FMI and major adverse cardiovascular events, we used Model 4, which included both Lean BMI and FMI for mutual adjustment. The median values of the sex-specific quartiles for Lean BMI and FMI were included in Models 1–4 as continuous variables for trend analysis [10].

To further explore the relationship between Lean BMI, FMI, and the primary endpoint and all-cause mortality in HFpEF patients, we conducted 4-knot restricted cubic splines analysis based on Model 4 to visualize these associations. Additionally, a two-segment linear regression model was used to determine the optimal cutoff point for Lean BMI. We divided the dataset into two groups based on this cutoff and calculated the HR and 95% confidence intervals for Lean BMI and FMI on either side of the cutoff. We then performed a log-likelihood ratio test to compare the single-line regression model with the two-segment model [10].

Interaction and subgroup analyses were conducted according to age [≤65 years vs. >65 years], sex, race, treatment group, smoking status, history of CAD, diabetes, COPD, hypertension, and hyperlipidemia. Sensitivity analyses were performed by excluding any major adverse cardiovascular events that occurred within the first 2 years of follow-up and using right-censoring for participants aged over 65. We also utilized different categories [third and fifth quintiles] to predict the associations between Lean BMI, FMI, and endpoint events.

All statistical analyses were two-sided, with a p-value of <0.05 considered statistically significant. Analyses were performed using Rstudio [version 2024.4.2.764].

## Results

A total of 3,445 participants were included in the TOPCAT trial. 125 participants were excluded due to the unavailability of data necessary to predict lean body mass or fat mass. The mean age of the included participants was 69 years [IQR 54–84], with 48.8% being male. Men had a higher Lean BMI compared to women, while women had a higher FMI: the mean Lean BMI in men was 20.28 kg/m² [IQR 16.43–24.13], and the mean FMI was 9.44 kg/m² [IQR 5.24–13.64]. In contrast, for women, the mean Lean BMI was 16.71 kg/m² [IQR 13.41–20.01], and the mean FMI was 14.1 kg/m² [IQR 8.08–20.12]. During the follow-up period, 624 participants [18.8%] experienced primary endpoint events, 503 participants (15.1%) experienced all-cause mortality, and 322 participants (9.7%) experienced cardiac death. Male participants were more likely to have a history of CAD, chronic obstructive pulmonary disease (COPD), diabetes mellitus (DM), atrial fibrillation (AF), smoking, and hyperlipidemia. Female participants were more likely to have a history of asthma and hypertension. The baseline characteristics of all included participants are detailed in Table 1.

When Lean BMI was included as a continuous covariate in the fully adjusted Cox proportional hazards model (Model 4), we did not find sufficient evidence for a statistically significant association between Lean BMI and primary endpoint (HR 0.98, 95% CI 0.82–1.16) (Table 2). However, it was associated with a reduction in the risk of all-cause mortality (HR 0.74, 95% CI 0.61–0.91) (Table 2). In contrast, when FMI was used as a continuous covariate in Model 4, each 1 standard deviation increase in FMI was associated with an increased risk of both the primary endpoint (HR 1.29, 95% CI 1.09–1.55) and all-cause mortality (HR 1.46, 95% CI 1.19–1.79). At the same time, we used BMI as the reference (model3). When BMI was used as the reference, every increase of 1 standard deviation would increase the occurrence of primary endpoint events (HR 1.21 95%CI 1.12–1.32), but it was not related to all-cause mortality (HR 1.07 95%CI 0.97–1.19). When we introduced waist circumference as a reference, the incidence of primary endpoint events and all-cause mortality increased with every 1 standard deviation increase in waist circumference (HR 1.24 95%CI 1.14–1.35) (HR 1.16 95%CI 0.1.05–1.28).

Table 3 present the association between the predicted Lean BMI and FMI with the primary endpoint. Q2 compared to Q1 is significant (HR 0.70, 95% CI 0.55–0.90), while others are not significant compared to Q1. In contrast, FMI was not independently associated with the primary event (HR 1.29, 95% CI 0.94–1.78).

Regarding all-cause mortality (Table 4), Lean BMI showed a significant negative association with the risk of all-cause mortality. Compared with participants in the Q1, patients in the Q2, Q3 and Q4 had decreased hazard ratios. Analysis of the relationship between FMI and all-cause mortality found that Q4 compared to Q1 was significant (HR 1.64, 95% CI 1.15–2.34), while others were not significant compared to Q1.

**Table 1. Baseline characteristics of included participants with HFpEF.**

| Characterisitc | Total (3320) | Male(1620) | Female(1700) | P value |
|---|---|---|---|---|
| Age | | | | |
| Median | 69 | 67 | 70 | |
| IQR | 54-84 | 52-82 | 56-84 | <0.05 |
| Sex(Male) | 1620(48.80) | | | |
| Race | | | | <0.05 |
| White(%) | 2603(78.40) | 1314(81.11) | 1289(75.82) | |
| Black(%) | 349(10.51) | 140(8.64) | 209(12.29) | |
| Other(%) | 368(11.08) | 166(10.25) | 202(11.88) | |
| Height, cm | | | | |
| Median | 167 | 174 | 160.02 | |
| IQR | 153-181 | 166-182 | 150.99-169.05 | <0.05 |
| Weight, cm | | | | |
| Median | 86 | 92 | 80 | |
| IQR | 60.8-112.2 | 65.69-118.31 | 56.56-103.44 | <0.05 |
| Waist, kg | | | | |
| Median | 104 | 106.68 | 100 | |
| IQR | 83.5-124.5 | 85.68-127.68 | 79.75-120.25 | <0.05 |
| BMI, kg/m$^2$ | | | | |
| Median | 30.8 | 30.38 | 31.23 | |
| IQR | 22.36-39.24 | 22.93-37.83 | 21.81-40.65 | <0.05 |
| Lean BMI, kg/m$^2$ | | | | |
| Median | 18.5 | 20.28 | 16.71 | |
| IQR | 14.05-22.95 | 16.43-24.13 | 13.41-20.01 | <0.05 |
| Fat percent(%) | | | | |
| Median | 39.7 | 31.71 | 45.39 | |
| IQR | 25.6-53.8 | 23.64-39.78 | 37.36-53.42 | <0.05 |
| Fat mass index, kg/m$^2$ | | | | |
| Median | 11.7 | 9.44 | 14.1 | |
| IQR | 5.63-17.77 | 5.24-13.64 | 8.08-20.12 | <0.05 |
| History | | | | |
| CAD(%) | 1978(59.58) | 1053(65) | 925(54.41) | <0.05 |
| Stroke(%) | 253(7.62) | 121(7.47) | 132(7.76) | 0.80 |
| COPD(%) | 375(11.30) | 229(14.14) | 146(8.59) | <0.05 |
| Asthma(%) | 205(6.17) | 74(4.57) | 131(7.71) | <0.05 |
| Hypertension(%) | 3040(91.57) | 1454(89.75) | 1586(93.29) | <0.05 |
| DM(%) | 1068(32.17) | 554(34.2) | 514(30.24) | <0.05 |
| AF(%) | 1174(35.36) | 632(39.01) | 542(31.88) | <0.05 |
| Smoker(%) | 352(10.60) | 260(16.05) | 92(5.41) | <0.05 |
| Hyperlipidemia(%) | 2001(60.27) | 1040(64.2) | 961(56.53) | <0.05 |
| Spironolactone(%) | 1664(50.12) | 809(49.94) | 855(50.29) | 0.86 |
| Primary endpoint(%) | 624(18.80) | 356(21.98) | 268(15.76) | <0.05 |
| All cause death(%) | 503(15.15) | 298(18.4) | 205(12.06) | <0.05 |
| Cardic death(%) | 322(9.70) | 191(11.79) | 131(7.71) | <0.05 |

BMI = body mass index; CAD = Coronary heart disease; COPD = Chronic obstructive pulmonary disease; DM = diabetes; AF = Atrial fibrillation; IQR = Inter-quartile range.

**Table 2. Hazard ratios (95% CI) for primary endpoint and all cause death, by predicted Lean BMI or FMI or BMI.**

**Hazard ratios (95% CI) for Primary endpoint, by predicted Lean BMI or FMI or BMI**

| Primary endpoint | model1* | P | model2† | P | model3‡ | P | model4§ | P |
|---|---|---|---|---|---|---|---|---|
| Lean BMI | 1.27(1.18-1.37) | < 0.05 | 1.35(1.25-1.47) | <0.05 | 1.20(1.09-1.32) | <0.05 | 0.98(0.82-1.16) | 0.79 |
| Fat mass index | 1.14(1.06-1.23) | < 0.05 | 1.45(1.33-1.58) | <0.05 | 1.27(1.15-1.40) | <0.05 | 1.30(1.09-1.55) | <0.05 |
| BMI | 1.24(1.15-1.33) | <0.05 | 1.36(1.26-1.47) | <0.05 | 1.21(1.12-1.32) | <0.05 | | |
| Waist | 1.37(1.27-1.48) | <0.05 | 1.40(1.30-1.52) | <0.05 | 1.24(1.14-1.35) | <0.05 | | |

**Hazard ratios (95% CI) for All cause death, by predicted Lean BMI or FMI or BMI**

| All cause death | model1* | P | model2† | P | model3‡ | P | model4§ | P |
|---|---|---|---|---|---|---|---|---|
| Lean BMI | 1.09(1.00-1.18) | 0.06 | 1.09(0.98-1.21) | 0.11 | 1.01(0.90-1.13) | 0.92 | 0.74(0.61-0.91) | <0.05 |
| Fat mass index | 0.96(0.88-1.06) | 0.44 | 1.24(1.11-1.38) | <0.05 | 1.14(1.02-1.28) | <0.05 | 1.46(1.19-1.79) | <0.05 |
| BMI | 1.02(0.93-1.12) | 0.65 | 1.15(1.05-1.26) | <0.05 | 1.07(0.97-1.19) | 0.18 | | |
| Waist | 1.22(1.12-1.33) | <0.05 | 1.25(1.14-1.37) | <0.05 | 1.16(1.05-1.28) | <0.05 | | |

BMI = body mass index, CI = confidence interval, HR = hazard ratio.

Unadjusted.

†Adjusted for age, sex and race.

‡Adjusted for age, sex, race, spironolactone treatment, history of CAD, hyperlipidemia, hypertension, smoking, diabetes, and COPD.

§Adjusted using characteristics for Model 3 by adding fat mass index or Lean BMI.

**Table 3. Hazard ratios (95% CI) for Primary endpoint, by Lean BMI and FMI quartiles.**

**Hazard ratios (95% CI) for Primary endpoint, by Lean BMI quartiles**

| Lean BMI quartiles | No. of events | incidence rate, % | HR (95%CI) | | | |
|---|---|---|---|---|---|---|
| | | | model1* | model2† | model3‡ | model4§ |
| 1st | 152 | 18.31 | Ref. | Ref. | Ref. | Ref. |
| 2nd | 117 | 14.09 | 0.70(0.55-0.89) | 0.77(0.60-0.98) | 0.76(0.60-0.97) | 0.70(0.55-0.90) |
| 3rd | 145 | 17.47 | 0.93(0.74-1.16) | 1.08(0.86-1.36) | 0.93(0.73-1.17) | 0.79(0.61-1.03) |
| 4th | 210 | 25.30 | 1.47(1.19-1.81) | 1.86(1.50-2.31) | 1.41(1.12-1.77) | 1.02(0.72-1.43) |
| trend P | | | 0.11 | 0.08 | 0.10 | 0.09 |

**Hazard ratios (95% CI) for Primary endpoint, by FMI quartiles**

| FMI quartiles | No. of events | incidence rate, % | HR (95%CI) | | | |
|---|---|---|---|---|---|---|
| | | | model1* | model2† | model3‡ | model4§ |
| 1st | 136 | 16.39 | Ref. | Ref. | Ref. | Ref. |
| 2nd | 126 | 15.18 | 0.91(0.71-1.16) | 0.93(0.73-1.19) | 0.87(0.68-1.11) | 0.85(0.66-1.08) |
| 3rd | 148 | 17.83 | 1.12(0.89-1.41) | 1.18(0.93-1.48) | 0.98(0.77-1.25) | 0.92(0.71-1.20) |
| 4th | 214 | 25.78 | 1.79(1.44-2.22) | 2.03(1.63-2.52) | 1.47(1.16-1.85) | 1.29(0.94-1.78) |
| trend P | | | 0.12 | 0.12 | 0.13 | 0.14 |

BMI = body mass index, FMI = Fat mass index, CI = confidence interval, HR = hazard ratio, Ref. = reference.

*Unadjusted.

†Adjusted for age, sex and race.

‡Adjusted for age, sex, race, spironolactone treatment, history of CAD, hyperlipidemia, hypertension, smoking, diabetes, and COPD.

§Adjusted using characteristics for Model 3 by adding Fat mass index or Lean BMI.

We also conducted a stratified analysis of the effect of spironolactone on the results of different body composition subgroups. The results showed that spironolactone had a protective effect on the primary endpoint only in the third quartile of lean body mass index. However, spironolactone had no effect on the results at the other subgroup levels (S2 Table in S1 File).

**Table 4. Hazard ratios (95% CI) for All cause death, by Lean BMI and FMI quartiles.**

**Hazard ratios (95% CI) for All cause death, by Lean BMI quartiles**

| Lean BMI quartiles | No. of events | incidence rate, % | HR (95%CI) | | | |
|---|---|---|---|---|---|---|
| | | | model1* | model2† | model3‡ | model4§ |
| 1st | 150 | 18.07 | Ref. | Ref. | Ref. | Ref. |
| 2nd | 111 | 13.37 | 0.66(0.51-0.84) | 0.73(0.57-0.94) | 0.74(0.58-0.95) | 0.66(0.51-0.86) |
| 3rd | 112 | 13.49 | 0.71(0.55-0.91) | 0.85(0.66-1.09) | 0.78(0.61-1.01) | 0.63(0.47-0.84) |
| 4th | 130 | 15.66 | 0.87(0.68-1.09) | 1.16(0.91-1.48) | 0.99(0.77-1.28) | 0.63(0.43-0.94) |
| trend P | | | 0.44 | 0.60 | 0.58 | 0.54 |

**Hazard ratios (95% CI) for All cause death, by FMI quartiles**

| FMI quartiles | No. of events | incidence rate, % | HR (95%CI) | | | |
|---|---|---|---|---|---|---|
| | | | model1* | model2† | model3‡ | model4§ |
| 1st | 133 | 16.02 | Ref. | Ref. | Ref. | Ref. |
| 2nd | 108 | 13.01 | 0.79(0.61-1.02) | 0.81(0.63-1.05) | 0.79(0.61-1.02) | 0.85(0.65-1.10) |
| 3rd | 110 | 13.25 | 0.83(0.65-1.07) | 0.88(0.69-1.14) | 0.80(0.61-1.03) | 0.92(0.69-1.23) |
| 4th | 152 | 18.31 | 1.25(0.99-1.58) | 1.49(1.17-1.88) | 1.25(0.97-1.61) | 1.64(1.15-2.34) |
| trend P | | | 0.29 | 0.30 | 0.27 | 0.29 |

BMI = body mass index, FMI = Fat mass index, CI = confidence interval, HR = hazard ratio, Ref. = reference.

*Unadjusted.

†Adjusted for age, sex and race.

‡Adjusted for age, sex, race, spironolactone treatment, history of CAD, hyperlipidemia, hypertension, smoking, diabetes, and COPD.

§Adjusted using characteristics for Model 3 by adding Fat mass index or Lean BMI.

In the restricted cubic spline analysis, the incidence of primary endpoint decreased sharply with an increase in Lean BMI, suggesting that extremely low Lean BMI (< 17.83 kg/m²) may be associated with a higher risk of primary endpoint events. However, once the Lean BMI reached a certain threshold, the influence on the incidence of endpoint events becomes trivial. (Fig 1A). There was a strong positive correlation between FMI and primary endpoint. When the FMI exceeded 7.75 kg/m², the risk of primary events increased significantly (Fig 1B).

For all-cause mortality (Fig 2A), Lean BMI showed a clear inverse association, with the risk of all-cause mortality decreasing until the Lean BMI exceeded 18.04 kg/m², where the curve flattened. Similarly, the effect of FMI on all-cause mortality followed a U-shaped pattern, showing a significant upward trend when it exceeded 7.72 kg/m² (Fig 2B).

We performed interaction and stratified analyses (Fig 3A–D). For the primary endpoint and all-cause mortality, there were significant interactions between age and Lean BMI and FMI, and increased Lean BMI and FMI were risk factors in the subgroup ≤70 years of age. In non-diabetic patients, Lean BMI and FMI were protective factors for the primary endpoint.

Our findings remained consistent in the sensitivity analyses, even after excluding primary endpoint and all-cause mortality in patients who experienced events within the first 2 years of follow-up or who were older than 70 years. The results also remained robust when using different categorizations of Lean BMI and FMI (thirds-group or fifths-group classifications) (S3-S10 Tables in S1 File).

## Discussion

Our analysis of body composition and its impact on prognosis in the HFpEF patient cohort indicates that a lower Lean BMI is associated with an increased risk of primary endpoint, while Lean BMI is inversely correlated with all-cause mortality. Conversely, FMI is positively correlated with both the primary endpoint and all-cause mortality.

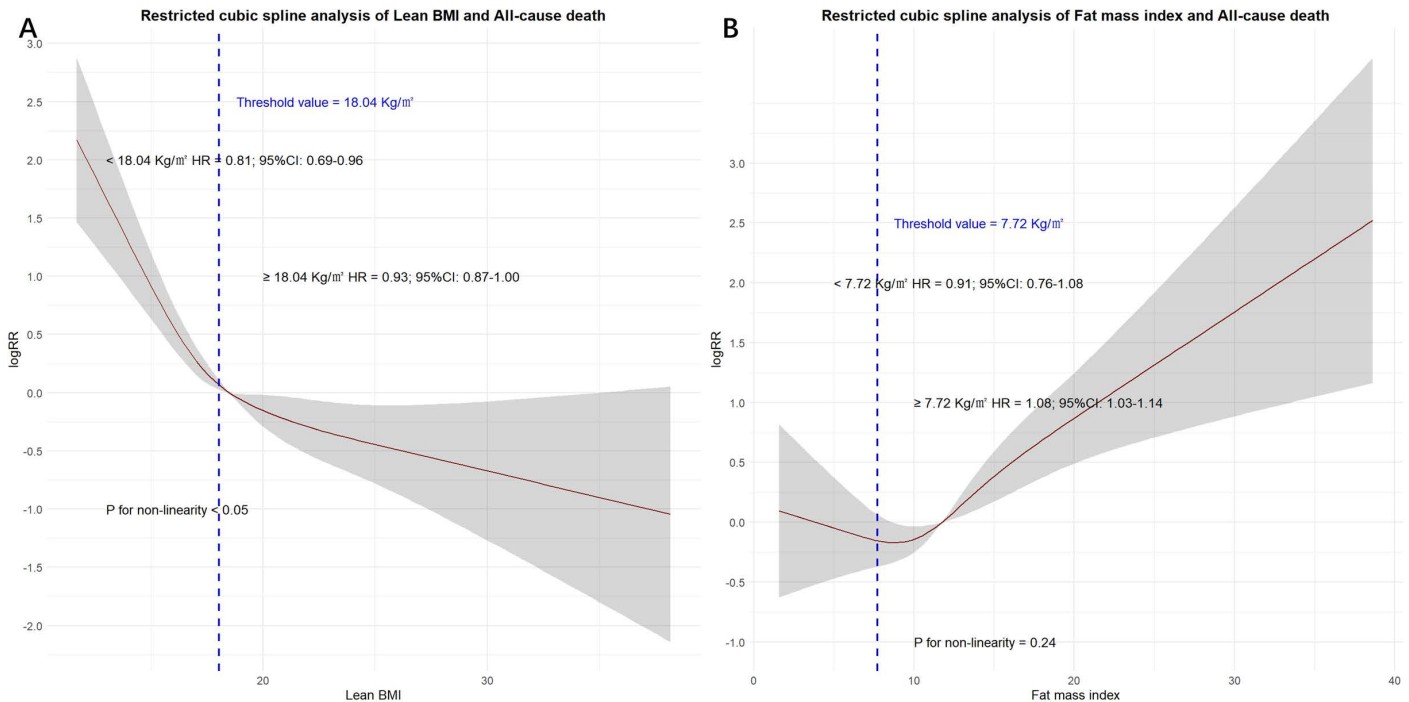

**Fig 1. Effect of Lean BMI and FMI on the primary endpoint.** (A) Lean body mass index. (B) Fat mass index. Red line denotes fitted curves and grey area represents 95% confidence intervals for the association between body mass index and adverse events. All models were adjusted for confounders in model 4.

In a previous analysis involving 69,273 HFpEF patients, the overall HR of BMI for all-cause mortality was 0.90 (95% CI 0.84–0.95), displaying a U-shaped correlation with the lowest risk point at 32–34 kg/m² [6]. However, BMI was positively correlated with heart failure hospitalizations. Earlier analyses of the TOPCAT US cohort also indicated a protective effect of BMI on all-cause mortality, and similar findings were observed in the HERO Trial cohort, demonstrating the "obesity paradox" [15]. Compared to patients with HFrEF, those with HFpEF are generally younger and have a higher fat percentage [11]. Our results indicate that patients with a higher FMI tend to have worse outcomes, and those with an excessively low Lean BMI also have poor prognoses. The same BMI may correspond to different body compositions, as individuals with varying levels of FMI and Lean BMI can have the same BMI. This heterogeneity in body composition contributes to the observed obesity paradox in HFpEF [6,10,13,16]. Thus, relying solely on BMI for prognostic evaluation in HFpEF may be inadequate. The study by Zhang et al. used many new obesity indicators to analyze the prognosis of HFpEF patients from the perspective of obesity, and also explained the impact of these new obesity indicators on all-cause mortality and heart failure hospitalization rates in HFpEF patients [17].

The U-shaped relationship between BMI and the primary endpoint and all-cause mortality can be further explained by the interactions between Lean BMI and FMI. We found that Lean BMI exerted a protective effect on primary endpoint events in patients with a Lean BMI below 17.83 kg/m² (and below 18.04 kg/m² for all-cause mortality). The increased risk in HFpEF patients with a lower BMI may result from the adverse effects of a low Lean BMI outweighing the beneficial effects of a lower FMI. When the protective influence of Lean BMI surpasses the harmful impact of FMI, an inverse relationship between BMI and prognosis emerges. Therefore, evaluating the effects of weight loss and prognosis solely based on BMI can be misleading. From the perspective of all-cause mortality, an increase in Lean BMI can effectively reduce the risk,

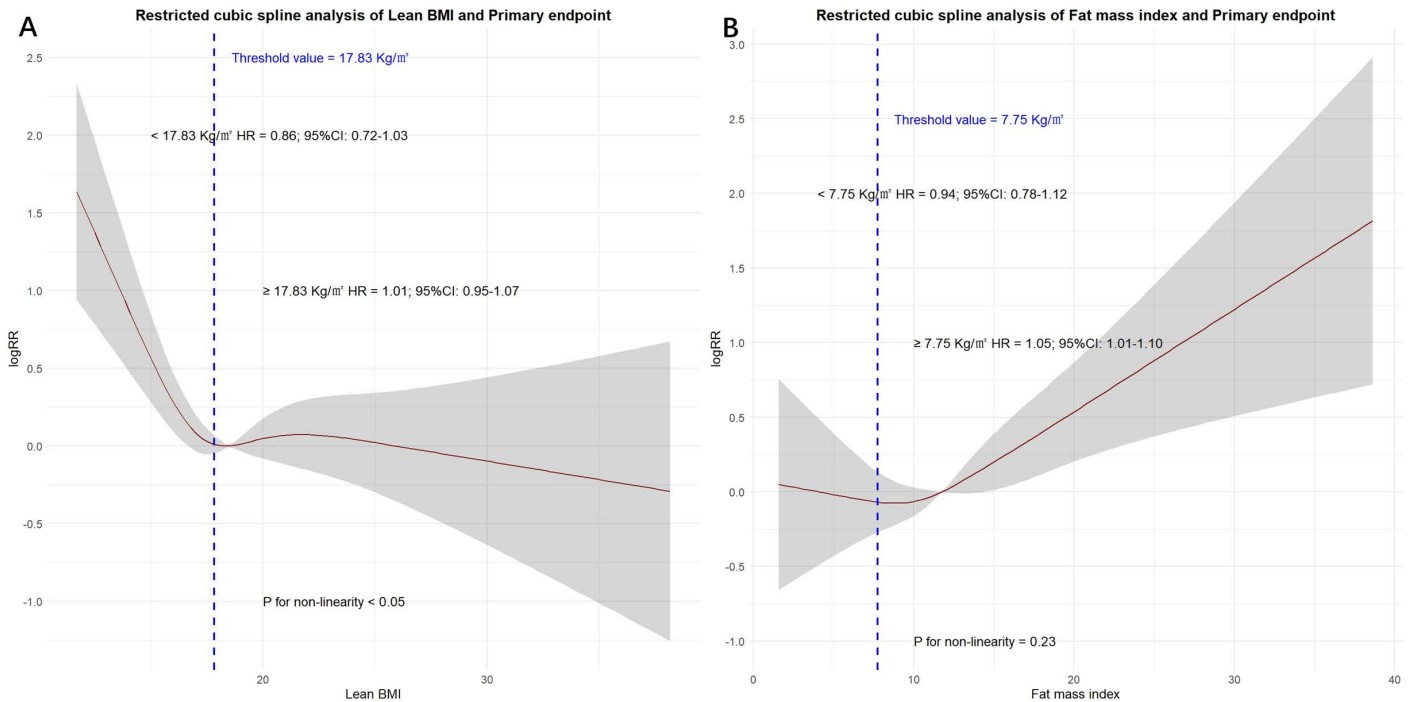

**Fig 2. Effect of Lean BMI and FMI on the All-cause death.** (A) Lean body mass index. (B) Fat mass index. Red line denotes fitted curves and grey area represents 95% confidence intervals for the association between body mass index and adverse events. All models were adjusted for confounders in model 4.

but its benefit plateaus beyond 17.83 kg/m² when considering major adverse events. Therefore, rather than merely maintaining BMI, it is more beneficial to preserve an adequate Lean BMI and reduce FMI to lower prognostic risk in patients.

Waist circumference is a commonly used obesity indicator, and its relationship with heart failure has been widely discussed [18]. An increase in waist circumference not only increases the risk of heart failure, but also affects the prognosis of heart failure [18–20]. In our study, waist circumference was included as a variable in the estimation formulas of Lean BMI and FMI. Therefore, studying Lean BMI and FMI may be more meaningful than evaluating waist circumference alone.

Previous research has shown that compared with HFrEF, HFpEF is more likely to result in abnormal body composition, leading to reduced cardiopulmonary capacity and consequently, a poor prognosis [2,3]. Lean BMI has been found to have a protective effect on prognosis in HF patients [16]. Among HFpEF patients, the loss of skeletal muscle mass and decline in skeletal muscle function are more pronounced than in HFrEF patients [3,21]. Improvements in cardiopulmonary function in HFpEF through exercise and other treatments are often due to enhanced peripheral, non-cardiac functions, consistent with our findings of a negative correlation between Lean BMI and prognosis. This result aligns with previous studies showing that sarcopenia is significantly associated with poor outcomes across all heart failure subtypes [12]. Sarcopenia, defined as the decline in muscle mass and function, has been widely recognized as a contributor to adverse symptoms and prognosis in heart failure patients [22,23]. It is crucial to recognize obese sarcopenia, a subtype in which patients may have a normal or high BMI but still exhibit the negative impacts of both sarcopenia and obesity [24]. This highlights the importance of a more detailed assessment of body composition and muscle function in evaluating patient prognosis.

Age is a common risk factor for HFpEF and affects the onset and prognosis of HFpEF [1]. Simultaneously, changes in body composition with age are also expected. Age is an important influencing factor when evaluating Lean BMI and Fat mass index [25]. In our interactive analysis and subgroup analysis, it is reasonable that age interacts with body

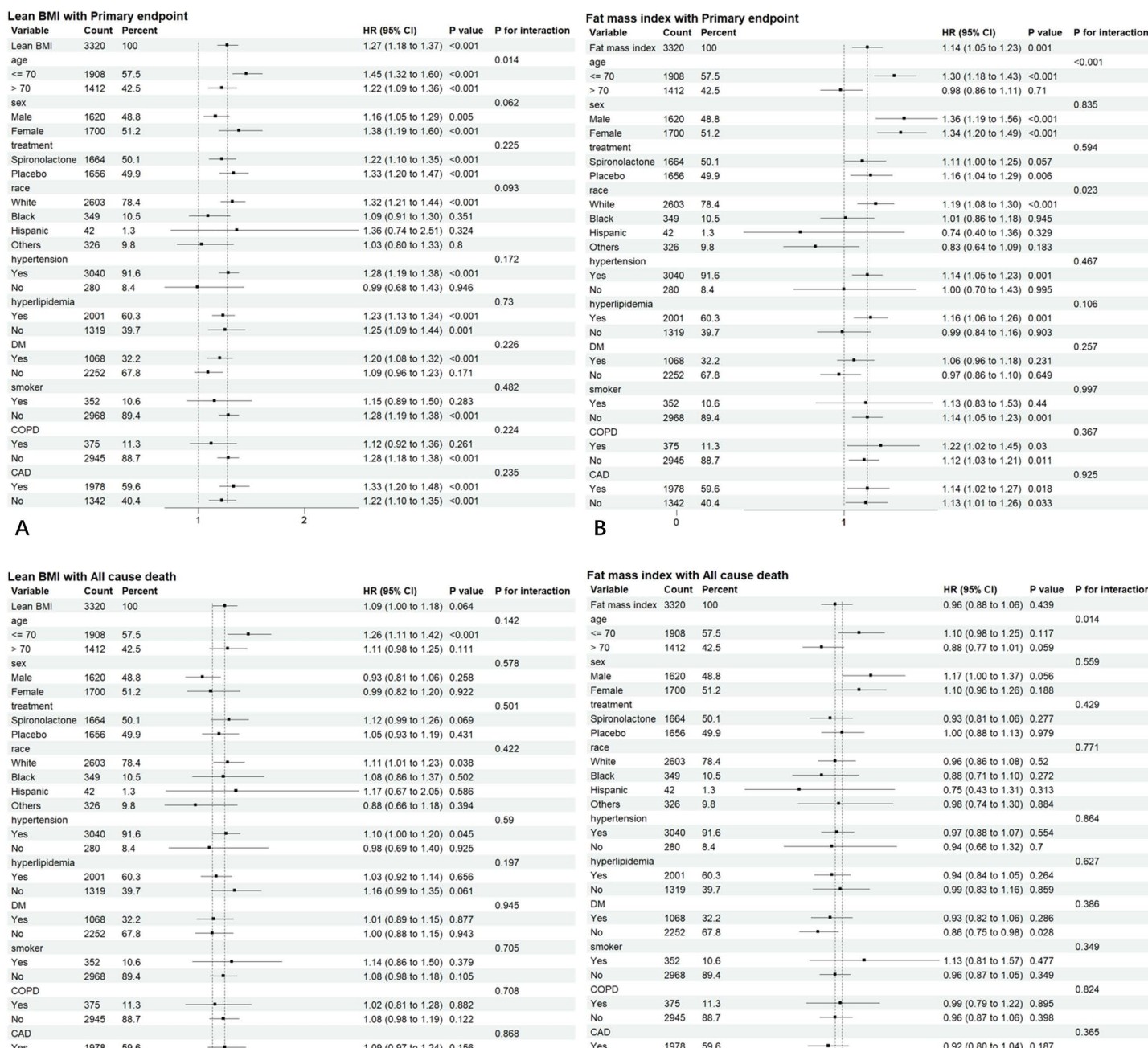

**Fig 3. A-B. Hazard ratios per 1 SD–increase in Lean BMI or FMI for Primary endpoint.** Each stratification was adjusted for all factors in model 4 (i.e., age, sex, race, spironolactone treatment, history of CAD, hyperlipidemia, hypertension, smoking, diabetes, COPD, Lean BMI or FMI) except for the stratification factor itself. Note: HR = hazard ratio, CI = confidence interval, Lean BMI = lean body mass index, FMI = fat mass index, CVD = cardiovascular disease, COPD = Chronic obstructive pulmonary disease, DM = diabetes. **C-D. Hazard ratios per 1 SD–increase in Lean BMI or FMI for All cause death.** Each stratification was adjusted for all factors in model 4 (i.e., age, sex, race, spironolactone treatment, history of CAD, hyperlipidemia, hypertension, smoking, diabetes, COPD, Lean BMI or FMI) except for the stratification factor itself. Note: HR = hazard ratio, CI = confidence interval, Lean BMI = lean body mass index, FMI = fat mass index, CVD = cardiovascular disease, COPD = Chronic obstructive pulmonary disease, DM = diabetes.

composition on the primary endpoint and all-cause mortality. Similarly, diabetes also changes the patient's muscle and fat metabolism, and a protective effect in non-diabetic patients is also expected [26,27].

In summary, it is not sufficient to use only BMI assessment of HFpEF patients. Many studies focus on obesity and explain the obesity paradox to a certain extent, but the analysis of muscle mass is often ignored. A comprehensive analysis of lean body mass and fat mass can better predict the patient's adverse prognosis and provide more targeted exercise advice to patients. Improving cardiac function and skeletal muscle metabolism in HFpEF patients through exercise is an effective way to improve the prognosis of HFpEF patients [28,29]. Reducing FMI through exercise while maintaining an appropriate lean BMI is a feasible method. Our results provide quantitative guidance for improving patient prognosis through exercise in the future.

### Limitations

Our study has several limitations. First, the Lean BMI and FMI obtained using the prediction equation are not perfect substitutes for actual measurements. Although previous research has demonstrated that this prediction method has high accuracy compared to DXA, it is not a direct measurement method [23]. Second, the data on weight and waist circumference were collected only at baseline, and changes during the follow-up period were not tracked. Third, this study utilized the complete dataset from TOPCAT. Prior studies have reported differences between the cohort outside North America and the North American cohort within the TOPCAT study, suggesting that the incidence rates in the non-North American cohort may be lower [30]. In recent years, the widespread use of sodium/glucose co-transporter 2 inhibitors (SGLT2i) and glucagon-like peptide-1 receptor agonist (GLP-1 RA) in patients with heart failure has effectively improved the prognosis of patients with heart failure, but this study did not involve them, and the effects of SGLT2i and GLP-1 RA were not effectively evaluated [31–33].

### Conclusions

In summary, In patients with HFpEF, a lower Lean BMI and a higher FMI were associated with an increased risk of primary endpoint events and all-cause mortality.

### Supporting information

**S1 File.** **S1 Table.** Prediction equations for Lean BMI, FMI and percent fat. **S2 Table.** Effects of spironolactone on body composition in different subgroups. **S3 Table.** Hazard ratio (95% CI) of Primary endpoint according to fifths of Lean BMI and FMI. **S4 Table.** Hazard ratio (95% CI) of All cause death according to fifths of Lean BMI and FMI. **S5 Table.** Hazard ratio (95% CI) of Primary endpoint according to thirds of Lean BMI and FMI. **S6 Table.** Hazard ratio (95% CI) of All cause death according to thirds of Lean BMI and FMI. **S7 Table.** Sensitivity analysis of Lean BMI and FMI in relation to Primary endpoint by excluding Primary endpoint occurred early in the first two years. **S8 Table.** Sensitivity analysis of Lean BMI and FMI in relation to All cause death by excluding All cause death occurred early in the first two years. **S9 Table.** Sensitivity analysis of Lean BMI and FMI in relation to Primary endpoint by excluding patients with age > 70 years. **S10 Table.** Sensitivity analysis of Lean BMI and FMI in relation to All cause death by excluding patients with age > 70 years.
(ZIP)

### Acknowledgments

Not applicable.
   **Consent for publication:** Not applicable.

## Author contributions

**Conceptualization:** Wen Su, Junyu Pei.

**Data curation:** Wen Su, Xiaopu Wang, Qian Wang, Junyu Pei.

**Formal analysis:** Wen Su, Xiaopu Wang, Qian Wang, Junyu Pei.

**Methodology:** Qian Wang, Junyu Pei.

**Resources:** Junyu Pei.

**Supervision:** Junyu Pei, Zhenfei Fang.

**Writing – original draft:** Wen Su, Junyu Pei.

**Writing – review & editing:** Zhenfei Fang.

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
