## [Decision Letter · Decision Letter 0]

5 Feb 2025

Dear Dr. pei,

Thank you for submitting your manuscript to PLOS ONE. After careful consideration, we feel that it has merit but does not fully meet PLOS ONE’s publication criteria as it currently stands. Therefore, we invite you to submit a revised version of the manuscript that addresses the points raised during the review process.

We look forward to receiving your revised manuscript.

Kind regards,

Amir Hossein Behnoush

Academic Editor

PLOS ONE

Journal Requirements:

[This study was supported by two major projects of Hunan Province, China: 2021SK2004; 2021SK1040;c].

Please confirm at this time whether or not your submission contains all raw data required to replicate the results of your study. Authors must share the “minimal data set” for their submission. PLOS defines the minimal data set to consist of the data required to replicate all study findings reported in the article, as well as related metadata and methods (https://journals.plos.org/plosone/s/data-availability#loc-minimal-data-set-definition ).

If your submission does not contain these data, please either upload them as Supporting Information files or deposit them to a stable, public repository and provide us with the relevant URLs, DOIs, or accession numbers. For a list of recommended repositories, please see https://journals.plos.org/plosone/s/recommended-repositories .

Reviewers' comments:

Reviewer's Responses to Questions

**Comments to the Author**

1. Is the manuscript technically sound, and do the data support the conclusions?

Reviewer #1: Yes

Reviewer #2: Yes

2. Has the statistical analysis been performed appropriately and rigorously?

Reviewer #1: I Don't Know

Reviewer #2: Yes

3. Have the authors made all data underlying the findings in their manuscript fully available?

Reviewer #1: Yes

Reviewer #2: Yes

4. Is the manuscript presented in an intelligible fashion and written in standard English?

Reviewer #1: Yes

Reviewer #2: Yes

Reviewer #1: Dear Authors, I had the privilege to review the manuscript PONE-D-24-54649 entitled "Association of predicted lean body mass and fat mass with prognosis in patients with heart failure preserved ejection fraction" for possibile publication in PLOS one.

In this trial you evaluated the role of lean BMI and FMI in a cohort of HF patients enrolled in the TOPCAT trial showing that a lower Lean BMI and a higher FMI were associated with an increased risk of primary endpoint events and all-cause mortality. The limits of a BMI only-centred evaluation in HF are already well known being the base of the "obesity paradox" concept, as you also explain in your manuscript.

Trying to estimate the body composition in HF patients is, therefore, important from a prognostic point of view.

My evaluation includes a positive feedback.

I suggest to expand the discussion section including more previous studies and trying to suggest more clinical implication. Moreover, could you analyze the impact of spironolactone in different lean BMI/FMI subgroups?

Reviewer #2: This manuscript is well-written, to-the-point, and statistically accurate. I have one point to recommend:

Several studies have shown that it is the waist circumference (and not BMI) that is a significant risk factor for many cardiovascular disorders and cardiovascular events. I believe at least some of these studies must be discussed in your manuscript since visceral fat, lean body mass, and fat mass are greatly emphasizing the waist circumference over BMI. Here are some articles that I recommend to add to your manuscript: PMID: 32020062 (discusses importance of waist circumference) ,PMID: 37227560 ( shows significant association between major cardiovascular events including heart failure and waist circumference, I highly recommend both)

**Do you want your identity to be public for this peer review?** For information about this choice, including consent withdrawal, please see our Privacy Policy

Reviewer #1: **Yes: ** Massimo Mapelli

Reviewer #2: **Yes: ** Alireza Ramandi

---

## [Author Response · Author response to Decision Letter 1]

3 Mar 2025

1. I suggest to expand the discussion section including more previous studies and trying to suggest more clinical implication. Moreover, could you analyze the impact of spironolactone in different lean BMI/FMI subgroups?

Thanks to your suggestions, we have supplemented the discussion section. We believe that the use of Lean BMI and FMI to evaluate the prognosis of HFpEF patients has positive significance. At the same time, these two indicators provide guidance for guiding patients to conduct exercise rehabilitation and body composition adjustment. At the same time, according to your suggestions, we have improved the analysis of the impact of spironolactone in Lean BMI and FMI for different subgroups, and the results are shown as supplementary material S2 Table. At the same time, we supplemented the use data of spironolactone in the study population in Table 1. I hope these contents can meet your expectations.

2. Several studies have shown that it is the waist circumference (and not BMI) that is a significant risk factor for many cardiovascular disorders and cardiovascular events. I believe at least some of these studies must be discussed in your manuscript since visceral fat, lean body mass, and fat mass are greatly emphasizing the waist circumference over BMI. Here are some articles that I recommend to add to your manuscript: PMID: 32020062 (discusses importance of waist circumference) ,PMID: 37227560 ( shows significant association between major cardiovascular events including heart failure and waist circumference, I highly recommend both)

Thank you for your suggestion. Waist circumference is indeed an important risk factor for many cardiovascular diseases and cardiovascular events. To this end, we show the effect of waist circumference on the primary endpoint and all cuase death of patients in Table 2. As you said, waist circumference is a very important factor. Lean BMI and FMI include waist circumference as an important indicator in the calculation process, which can also partially reflect the important position of waist circumference in our study. Regarding the reference PMID: 32020062 you recommended, we appreciate your help and have included it in our references, which is very helpful for our discussion. Regarding the reference PMID: 37227560, unfortunately, we read this article and found it not relevant to our research, so we did not include it. However, your caption inspired us, and we included more content mentioned in your caption and discussed it, and also added relevant references. Thank you very much for your suggestions, and I hope these contents can meet your expectations.

---

## [Decision Letter · Decision Letter 1]

11 Apr 2025

Association of predicted lean body mass and fat mass with prognosis in patients with heart failure preserved ejection fraction

PONE-D-24-54649R1

Dear Dr. pei,

We’re pleased to inform you that your manuscript has been judged scientifically suitable for publication and will be formally accepted for publication once it meets all outstanding technical requirements.

Kind regards,

Amir Hossein Behnoush

Academic Editor

PLOS ONE

Additional Editor Comments (optional):

Reviewers' comments:

Reviewer's Responses to Questions

**Comments to the Author**

Reviewer #1: All comments have been addressed

Reviewer #2: All comments have been addressed

2. Is the manuscript technically sound, and do the data support the conclusions?

Reviewer #1: Yes

Reviewer #2: Yes

3. Has the statistical analysis been performed appropriately and rigorously?

Reviewer #1: I Don't Know

Reviewer #2: Yes

4. Have the authors made all data underlying the findings in their manuscript fully available?

Reviewer #1: Yes

Reviewer #2: Yes

5. Is the manuscript presented in an intelligible fashion and written in standard English?

Reviewer #1: Yes

Reviewer #2: Yes

Reviewer #1: I think the article improved after the authors addressed the minor revisions requested. The discussion has been expanded accordingly

Reviewer #2: thank you for your revision. All comments have been addressed and all questions have been addressed appropriately.

**Do you want your identity to be public for this peer review?** For information about this choice, including consent withdrawal, please see our Privacy Policy

Reviewer #1: **Yes: ** Massimo Mapelli

Reviewer #2: **Yes: ** Alireza Ramandi

---

## [Editor Report · Acceptance letter]

PONE-D-24-54649R1

PLOS ONE

Dear Dr. pei,

I'm pleased to inform you that your manuscript has been deemed suitable for publication in PLOS ONE. Congratulations! Your manuscript is now being handed over to our production team.

Kind regards,

on behalf of

Dr. Amir Hossein Behnoush

Academic Editor

PLOS ONE